# Maximizing Revenue under Market Shrinkage and Market Uncertainty

**Maria-Florina Balcan**
School of Computer Science
Carnegie Mellon University
ninamf@cs.cmu.edu

**Siddharth Prasad**
Computer Science Department
Carnegie Mellon University
sprasad2@cs.cmu.edu

**Tuomas Sandholm**
Computer Science Department
Carnegie Mellon University
Optimized Markets, Inc.
Strategic Machine, Inc.
Strategy Robot, Inc.
sandholm@cs.cmu.edu

## Abstract

A shrinking market is a ubiquitous challenge faced by various industries. In this paper we formulate the first formal model of shrinking markets in multi-item settings, and study how mechanism design and machine learning can help preserve revenue in an uncertain, shrinking market. Via a sample-based learning mechanism, we prove the first guarantees on how much revenue can be preserved by truthful multi-item, multi-bidder auctions (for limited supply) when only a random unknown fraction of the population participates in the market. We first present a general reduction that converts any sufficiently rich auction class into a randomized auction robust to market shrinkage. Our main technique is a novel combinatorial construction called a *winner diagram* that concisely represents all possible executions of an auction on an uncertain set of bidders. Via a probabilistic analysis of winner diagrams, we derive a general possibility result: a sufficiently rich class of auctions always contains an auction that is robust to market shrinkage and market uncertainty. Our result has applications to important practically-constrained settings such as auctions with a limited number of winners. We then show how to efficiently learn an auction that is robust to market shrinkage by leveraging practically-efficient routines for solving the winner determination problem.

## 1 Introduction

A shrinking market with uncertain buyer participation is a natural phase of products' and services' lifecycles. Current examples of great importance include media consumers—known as cord cutters—who cancel cable-TV subscriptions in favor of streaming services [29], a thinning customer base for department stores due to online retailers like Amazon [11, 16], and reduced capacities for restaurants during the COVID-19 pandemic [43]. In this paper we study how mechanism design can help preserve revenue in this ubiquitous challenge of a shrinking market, specifically for combinatorial auctions for limited supply. The seller has $m$ indivisible items to allocate to a set $S$ of $n$ bidders. The bidders can express how much they value each possible bundle $b \subseteq \{1, \ldots, m\}$ of items. Combinatorial auctions have had wide reach in practice, from strategic sourcing to spectrum auctions to estate auctions. Cramton et al. [10] provide a survey of various aspects of combinatorial auctions. The design of revenue-maximizing combinatorial auctions in multi-item, multi-bidder settings is an elusive and difficult problem that has spurred a long and active line of research combining techniques from economics, artificial intelligence, and theoretical computer science. This is still largely an open question and a very active research area.

36th Conference on Neural Information Processing Systems (NeurIPS 2022).

In this paper we start a new strand within that topic area, namely the study of shrinking markets with uncertain buyer participation. We introduce the first formal model of market shrinkage in multi-item settings, and prove the first revenue guarantees. Specifically, we show how much revenue can be preserved when only a random unknown fraction of the set $S$ of bidders participates in the market.

## 1.1  Summary of the contributions of this paper

We present the first formal analysis of how much revenue can be preserved in a shrinking market, for multi-item settings. More precisely, there is a set $S$ of $n$ bidders that is known to the mechanism designer. Each bidder participates in the market independently with probability $p$, but the valuations of the bidders who participate in the market, denoted by $S_0 \subseteq S$, are unknown (what is known is that they belong to $S$). We present a learning-based method for designing a mechanism that satisfies the first known revenue-preservation guarantees in this setting.

In Section 2 we provide a formal description of the problem setting, including formal definitions of the auction settings we consider. We precisely show how to reckon with subtleties that arise when auctions are run among a shrunken market of unknown size.

In Section 3 we provide and discuss a simple example of a market where reduced competition in a shrinking market drives revenue to a lower threshold than one might expect. We furthermore show that if bidders's valuation functions can depend on what items other bidders receive, there exist scenarios in which only an exponentially small (in the number of items) fraction of the revenue obtainable by even the vanilla Vickrey-Clarke-Groves (VCG) auction on $S$ can be guaranteed on a random subset of bidders, even if a large fraction of the market shows up. For example, if 50 items are for sale and each bidder shows up independently with 90% probability, our construction yields a maximum expected revenue of roughly 7% of the VCG revenue on $S$. If 100 items are for sale, at most 0.52% of the VCG revenue on $S$ can be guaranteed.

Our main theorem is the following revenue guarantee obtained via a sample-based learning algorithm. Delineability is a structural assumption introduced by Balcan et al. [3] satisfied by nearly all commonly studied auction classes. $W_{\mathcal{M}}(S)$ denotes the maximum welfare achievable by mechanisms in $\mathcal{M}$, $k$ is a term that depends on the number of winners in any mechanism in $\mathcal{M}$, and $\gamma$ is a constant that depends on $S$. $\mathsf{Rev}_M$ denotes the revenue function induced by $M$. All notation and formal definitions are in the following sections. Sections 4 and 5 are dedicated to proving Theorem 1.1.

**Theorem 1.1.** *Let $\mathcal{M}$ be $(d, h)$-delineable class of mechanisms. A mechanism $\widetilde{M} \in \mathcal{M}$ such that*

$$\mathbb{E}[\mathsf{Rev}_{\widetilde{M}}(S_0)] \geq \Omega\left(\frac{p^2}{k^{1+\log_{1/\gamma}(4/p)}}\right) W_{\mathcal{M}}(S) - \varepsilon$$

*with probability at least $1 - \delta$ can be computed in $NhT + (Nh)^{O(d)}$ time, where $T$ is the time required to generate any given hyperplane witnessing delineability of any mechanism in $\mathcal{M}$ and $N = O\left(\frac{d \log(dh)}{\varepsilon^2} \log(\frac{1}{\delta})\right)$.*

In Section 4 we prove that $\sup_{M \in \mathcal{M}} \mathbb{E}[\mathsf{Rev}_M(S_0)] \geq \Omega(\frac{p^2}{k^{1+\log_{1/\gamma}(4/p)}}) W_{\mathcal{M}}(S)$, which is the major technical contribution of this paper. Our main technique is the analysis of a novel combinatorial structure we construct called a *winner diagram*, which is a graph that concisely captures all possible executions of an auction on an uncertain set of bidders. Via a probabilistic method argument that randomizes over a subgraph of the winner diagram, we arrive at a general possibility result: *if $\mathcal{M}$ is a sufficiently rich class of mechanisms, there always exists an $M \in \mathcal{M}$ that is robust to uncertainty/shrinkage in the market*. This implies our bound on $\sup \mathbb{E}[\mathsf{Rev}_M(S_0)]$. We primarily focus on the case where bidders participate in the market independently with probability $p$, but show how to generalize our results to any distribution over submarkets. Our bound is a parameterized guarantee that has interesting applications to practically motivated auction constraints: (1) limiting the number of winners and (2) bundling constraints on items.

In Section 5 we present a learning algorithm to compute a mechanism $\widetilde{M}$ such that $\mathbb{E}[\mathsf{Rev}_{\widetilde{M}}(S_0)] \geq \sup_M \mathbb{E}[\mathsf{Rev}_M(S_0)] - \varepsilon$ with high probability, which proves Theorem 1.1. Our algorithm exploits geometric structure and a linear-programming approach over hyperplane arrangements. We show the run-time of our procedure is computationally tractable for a specific auction class by leveraging practically-efficient routines for solving the winner determination problem.

## 1.2 Related work

**Shrinking markets and uncertainty**  Shrinking markets have been studied by various researchers in the context of oil companies [47], cable TV [1, 29], labor markets [23], telecom markets [31], housing markets [24], and in combinatorial settings including a thinning customer base for department stores due to online retailers like Amazon [11, 16] and reduced capacities for restaurants during the COVID-19 pandemic [43]. Most of this existing research is extremely domain specific, and provides advisory content based on historical observations, data, and general economic knowledge. We introduce the first formal model of market shrinkage in multi-item settings, and prove the first known guarantees for how much revenue can be preserved in a shrinking market. Our guarantee on the revenue preserved in a shrinking market provides a positive contrast to recent work of Dobzinski and Uziely [12], who study the effect of market shrinkage on revenue loss. They show that even in the case of selling a single item to $n$ buyers with known valuation distributions, the absence of a single buyer with a fixed "low" value can surprisingly result in a (multiplicative) revenue loss of $\frac{1}{e+1}$ (in expectation). We tackle the significantly more complex multi-item setting. Furthermore, our main results are prior-free (in that they are tailored to the specific set $S$ of bidders and do not require bidders to come from a distribution) and thus provide a strong positive contrast to this negative result.

Our results can also be viewed from the perspective of an uncertain market, since at the point of the mechanism design the subset of bidders that participates in the market is unknown. Mechanism design with uncertainty about bidder valuations has previously studied [28, 45], but to the best of our knowledge the prior-free setting for combinatorial auctions has not been considered.

**Revenue in combinatorial auctions**  We prove guarantees when the seller limits the number of winners and when the seller places bundling constraints on the items (for example, by enforcing that certain items must be sold together). Reasons for limiting the number of winners include: (1) avoiding the logistical hassle of having a large number of winners – this constraint is commonly used in sourcing auctions [21, 36, 37, 41] and (2) increasing competition to boost revenue, as is studied by Roughgarden et al. [34] (though in a different setting than ours). Kroer and Sandholm [25] show that the VCG auction run with bundling constraints can yield significant revenue gains.

Furthermore, catalyzed by the seminal work of Bulow and Klemperer [7], a recent line of work studies the competition complexity of auctions, and provides results that compare the optimal (expected) revenue to the revenue of mechanisms like VCG when the number of bidders is augmented [6, 8, 14, 15, 34]. Competition complexity results can be seen as tackling the "opposite" situation of a growing market. However, existing work in this area uses the very different objective of expected revenue over buyer valuation distributions, and moreover has only tackled restrictive classes of valuation functions (such as unit-demand and additive valuations). Also related is the work of Rastegari et al. [32], who study settings where revenue can counterintuitively be boosted by dropping bidders.

**Learning for auction design**  Our algorithm for designing a mechanism robust to uncertain market shrinkage is a sample-based learning algorithm. Such *sample-based automated mechanism design* methods were introduced by Sandholm and Likhodedov [26, 27, 42], with the first generalization guarantees for machine learning based mechanism design appearing in Balcan et al. [2]. Machine learning for auction design has since grown into a rich field blending theory and practice [3, 4, 13, 30]. Unlike most prior work on machine learning for auction design, our setting is prior-free and does not have access to samples. In our learning algorithm, the mechanism designer produces samples himself that simulate the shrunken market, which is along the lines of the learning-within-an-instance paradigm of Balcan et al. [5]. For simplicity, we use the term *sample-based* as a catch-all for any algorithm that uses samples, prior-free or not. Furthermore, while our algorithm is a sample-based, the techniques we use to prove our main guarantee are fundamentally new. The aforementioned literature seeks to find a near-optimal mechanism from samples—in contrast our goal is to prove concrete revenue guarantees on the optimal mechanism.

## 2 Problem formulation

**Combinatorial auctions**  A seller has $m$ indivisible items to allocate among a set $S$ of $n$ bidders. Each bidder is described by her combinatorial valuation function $v_i : 2^{\{1,\ldots,m\}} \to \mathbb{R}_{\geq 0}$. We assume

bidders' valuations satisfy *free-disposal*, that is, $v_i(b) \le v_i(b')$ for all $b \subseteq b' \subseteq \{1, \ldots, m\}$, and $v_i(\emptyset) = 0$. For an allocation $\alpha$, $v_i(\alpha)$ is bidder $i$'s value for the bundle she receives under $\alpha$.

A mechanism $M$ specifies, given a set of bidders $S$, an allocation $\alpha$ of the items and a payment $p_i$ to be collected from each bidder $i \in S$. The welfare of $M$ when run on $S$ is $W_M(S) = \sum_{i \in S} v_i(\alpha)$, and its revenue on $S$ is $\mathsf{Rev}_M(S) = \sum_{i \in S} p_i$. $M$ is *incentive compatible* if no bidder $i$ may strictly increase her utility (value minus payment) by misreporting her true valuation function $v_i$ to $M$. $M$ is *individually rational* if $v_i(\alpha) \ge p_i$ for all $i$. All mechanisms considered in this paper are assumed to be incentive compatible and individually rational. The classical Vickrey-Clarke-Groves (VCG) auction [9, 17, 48] uses an efficient allocation $\alpha^*$ (one that maximizes $\sum_{i \in S} v_i(\alpha)$), and charges bidder $i$ a payment of $\max_\alpha \sum_{j \ne i} v_j(\alpha) - \sum_{j \ne i} v_j(\alpha^*)$. The VCG auction is truthful and individually rational. It is well known that the VCG auction can yield low revenue in many scenarios.

**Market-size uncertainty**    In our model, the mechanism designer has full knowledge of the entire population of bidders $S$ (described by their valuation functions). An unknown random subset of $S$ participates in the market. We write $S_0 \sim_p S$ to denote a subset $S_0$ that is sampled from $S$ by including each bidder in $S_0$ independently with probability $p$. More generally, for a distribution $D$ over $2^S$, we write $S_0 \sim_D S$ to denote a random subset of $S$ chosen according to $D$. We are interested in what happens to the maximum revenue achievable when only a random fraction of the set $S$ of bidders participates in the auction, that is, $\sup_{M \in \mathcal{M}} \mathbb{E}_{S_0 \sim_D S}[\mathsf{Rev}_M(S_0)]$.

Since a variable group of bidders of variable size can participate in the auction mechanisms we run, we require the important assumption that auctions in $\mathcal{M}$ can be run on variable-size sets of bidders in a well-defined manner. Various well-studied classes of auctions satisfy this property: examples include the class of VCG auctions with reserve prices, $\lambda$-auctions [22], affine-maximizer auctions [33], and various population-size-independent auctions in Balcan et al. [5]. We assume that the mechanism designer knows the valuations of the bidders in $S$ to begin with. So, each bidder can be thought of having an identity (for example, "the bidder who values apples at $x$ and oranges at $y$", or "the bidder with valuation function $v_4$"), and the mechanism designer knows the identities/valuations $v_1, \ldots, v_n$ of all bidders in $S$. An allocation, formally, is a mapping from items to bidder identities.

The sequence of mechanism design and revelation in our setting is the same as in the standard mechanism design setting. Specifically, the mechanism design (computation of a mechanism from $\mathcal{M}$) takes place before the bidders in the shrunken market are asked to reveal their valuations. This is important for incentive compatibility, that is, for motivating the bidders to reveal their true valuations. If the design/choice of $M \in \mathcal{M}$ is allowed to be based on the revealed valuations, the auction might not be incentive compatible. (The class of second price auctions with reserve prices for a single item serves as an illustrative example. Choosing the reserve price to maximize revenue after the shrunken market is revealed clearly violates incentive compatibility.) Because the designer does not know exactly which bidders are in the shrunken market $S_0$, the designer has uncertainty about the valuations of the bidders. He only knows that they belong to $S$.

**Assumptions on $\mathcal{M}$ and $S$**    For any $S' \subseteq S$, let $W_\mathcal{M}(S') = \max_{M \in \mathcal{M}} W_M(S')$ and $\mathsf{Rev}_\mathcal{M}(S') = \max_{M \in \mathcal{M}} \mathsf{Rev}_M(S')$. Let $\mathsf{win}_M(S')$ denote the set of bidders in $S'$ that win a nonempty bundle of items per $M$. Let $\mathsf{win}_\mathcal{M}(S')$ denote the set of bidders in $S'$ that win a nonempty bundle of items per the mechanism in $\mathcal{M}$ achieving $W_\mathcal{M}(S')$. The following two assumptions are the most critical ones.

**Welfare submodularity**  For all $S_1, S_2 \subseteq S$, $W_\mathcal{M}(S_1) + W_\mathcal{M}(S_2) \ge W_\mathcal{M}(S_1 \cup S_2) + W_\mathcal{M}(S_1 \cap S_2)$.
**Winner monotonicity**  For all $S'' \subseteq S' \subseteq S$ and all $i \in S''$, $i \in \mathsf{win}_\mathcal{M}(S') \implies i \in \mathsf{win}_\mathcal{M}(S'')$.

Suppose $W_\mathcal{M}(S') = W_{VCG}(S')$ for any $S' \subseteq S$, that is, $\mathcal{M}$ is sufficiently rich to be able to allocate items efficiently (as is the case with all mechanisms in the hierarchies discussed by Balcan et al. [3, 5]). Then, welfare submodularity implies winner monotonicity [19]. If the valuation functions of bidders in $S$ satisfy the gross-substitutes property, then both welfare submodularity and winner monotonicity hold [18, 19, 49].

The final assumptions stipulate that $\mathcal{M}$ is a sufficiently rich class of mechanisms. We assume that $\mathsf{Rev}_\mathcal{M}(S') = W_\mathcal{M}(S') = W_{VCG}(S')$ for all $S' \subseteq S$, and further that $\mathcal{M}$ satisfies the following "global VCG-like" property: $\mathsf{Rev}_\mathcal{M}(S')$ depends only on $\mathsf{win}_\mathcal{M}(S')$ and $\mathsf{win}_\mathcal{M}(S' \setminus \{i\})$ for each $i \in \mathsf{win}_\mathcal{M}(S')$. In words, these conditions stipulate the following: (1) $\mathcal{M}$ is sufficiently rich such that in a non-truthful full-information setting, $\mathcal{M}$ can always extract the full social surplus

$W_{\mathcal{M}}(S') = W_{VCG}(S')$ as revenue and (2) for any $S'$, the payments collected by the revenue-maximizing mechanism $M$ that achieves $\mathsf{Rev}_{\mathcal{M}}(S')$ depend only on $W_{\mathcal{M}}(S)$ and the maximum welfares $W_{\mathcal{M}}(S \setminus \{i\})$ achievable when each bidder drops out.

As a concrete example, if $S$ is a set of bidders with gross-substitutes valuations, then the class of $\lambda$-auctions and the class of affine-maximizer auctions satisfy all of the above properties (we prove this fact in Appendix A, and provide more details about these auction classes).

## 3 Revenue loss can be drastic

At first glance it might appear that the expected revenue preserved by a mechanism $M$ when each bidder participates independently with probability $p$ should simply be $p \cdot \mathsf{Rev}_M(S)$ (or more if one thinks of revenue as having diminishing returns in the number of bidders). This intuition is indeed accurate if $\mathsf{Rev}_M$ is a submodular function (which captures the diminishing returns property). However, revenue can shrink by more than this when mechanisms in $\mathcal{M}$ do not have submodular revenue. One reason for greater revenue loss is reduced competition among buyers. For example, suppose there are $m$ items and $2m$ bidders, where bidder $i$ for $1 \le i \le m$ has valuation $v_i(b) = c$ if $i \in b$ and $v_i(b) = 0$ otherwise, and bidder $m + i$ for $1 \le i \le m$ has valuation $v_{m+i}(b) = c - \varepsilon/m$ if $i \in b$ and $v_{m+i}(b) = 0$ otherwise (bidders have combinatorial valuations in this example, so valuation functions only depend on the bundle of items received). The VCG auction will allocate item $i \in \{1, \ldots, m\}$ to bidder $i$. The payment collected from bidder $i$ will be $c - \varepsilon/m$, which is the second highest value for item $i$. The revenue from VCG is thus $mc - \varepsilon = W(S) - \varepsilon$. Now, suppose each bidder participates in the auction independently with probability $p$. The expected revenue can be computed by breaking it up across items:

$$\mathbb{E}[\mathsf{Rev}_{VCG}(S_0)] = \sum_{i=1}^{m} \mathbb{E}[\text{Revenue from item } i] = \sum_{i=1}^{m} p^2(c - \varepsilon/m) = p^2(W(S) - \varepsilon).$$

The third equality is due to the fact that VCG generates nonzero revenue from item $i$ if and only if both bidders $i$ and $m + i$ participate, since if at most one of them shows up there is no competition for that item. So $\mathbb{E}[\mathsf{Rev}_{VCG}(S_0)] = p^2 \cdot \mathsf{Rev}_{VCG}(S)$.

Furthermore, if bidders' valuations are *allocational*, that is, $v_i(\alpha)$ can depend on what items other bidders receive, revenue loss can be even more dramatic.

**Theorem 3.1.** *For any $\varepsilon > 0$ there exists a set $S$ of bidders with allocational valuations such that for $S_0 \sim_p S$, $\mathbb{E}[\mathsf{Rev}_M(S_0)] \le p^{m/2} \cdot (\mathsf{Rev}_{VCG}(S) + 2\varepsilon) + \varepsilon$ for any individually rational auction $M$.*

The exponential revenue decay in the number of items means that even if the shrunken market is large in expectation, the revenue loss can be dramatic. For example, if 50 items are for sale and each bidder shows up independently with 90% probability, our construction shows that any auction can guarantee only at most roughly 7% of the VCG revenue on $S$. If 100 items are for sale, at most 0.52% of the VCG revenue on $S$ can be guaranteed.

## 4 Main guarantee on preserved revenue

We now present our main revenue guarantee when each bidder participates in the auction independently with probability $p$. For a set of bidders $S' \subseteq S$, let $\omega(S') = \mathsf{win}_{\mathcal{M}}(S') \cup (\cup_{i \in S'}\mathsf{win}_{\mathcal{M}}(S'\setminus\{i\}))$ be the set of bidders in $S'$ whose valuations determine $\mathsf{Rev}_{\mathcal{M}}(S')$. Define an equivalence relation $\equiv$ on subsets of $S$ by $S_1 \equiv S_2$ if and only if $\omega(S_1) = \omega(S_2)$. Let $\varphi(S') = \frac{1}{n}\sum_{i=1}^{n} W_{\mathcal{M}}(S' \setminus \{i\})$. $\varphi$ serves as a potential function in the proof of the following theorem and represents the average max-welfare of $S'$ when a uniformly random bidder in $S$ drops out.

---

Mechanism $\mathcal{A}$
(1) Let $S_1, \ldots, S_\ell$ be an enumeration of the equivalence classes with $\varphi(S_i) > \frac{p}{4}W_{\mathcal{M}}(S)$.
(2) Let $M_1, \ldots, M_\ell$ denote the mechanisms that achieve $\mathsf{Rev}_{\mathcal{M}}(S_1), \ldots, \mathsf{Rev}_{\mathcal{M}}(S_\ell)$.
(3) Choose $M$ uniformly at random from $\{M_1, \ldots, M_\ell\}$, and run $M$.

---

The main challenge in analyzing this mechanism is bounding $\ell$. Before we do that, we analyze the revenue guarantee it satisfies in terms of $\ell$.

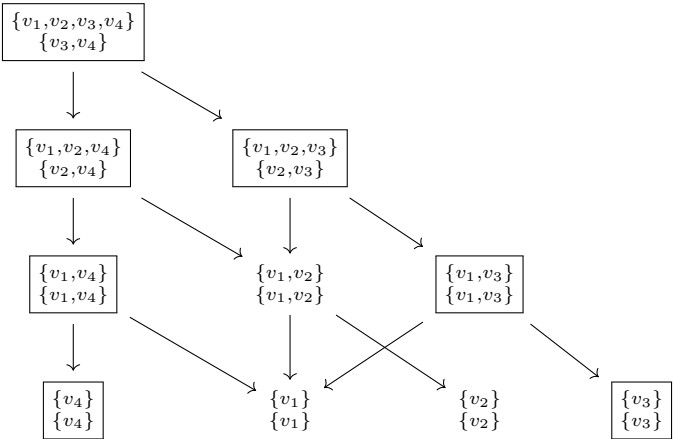

Figure 1: A winner diagram representing a second-price auction with a single item and four bidders with valuations $S = \{v_1 = 1, v_2 = 2, v_3 = 4, v_4 = 8\}$. At each node, the top set $S'$ is the set of remaining bidders, and the bottom set is the set of bidders $\omega(S')$ that actually determine revenue. Boxed nodes represent heavy equivalence classes for $p = 8/9$, which is the subgraph of the winner diagram $\mathcal{A}$ randomizes over.

**Lemma 4.1.** *Let $|S| \geq 2$. For $S_0 \sim_p S$, $\mathbb{E}[\mathsf{Rev}_{\mathcal{A}}(S_0)] \geq \Omega(p^2/\ell)W_{\mathcal{M}}(S)$.*

*Proof.* By definition of $\omega$, if $S_1 \equiv S_2$, then $W_{\mathcal{M}}(S_1) = W_{\mathcal{M}}(S_2)$, $\varphi(S_1) = \varphi(S_2)$, and $\mathsf{Rev}_{\mathcal{M}}(S_1) = \mathsf{Rev}_{\mathcal{M}}(S_2)$ (and the maximum revenue is achieved by the same $M \in \mathcal{M}$ for both sets). Call a set of bidders $S' \subseteq S$ *heavy* if $\varphi(S') > \frac{p}{4}W_{\mathcal{M}}(S)$. If $S_0$ is heavy, there is $M \in \{M_1, \ldots, M_\ell\}$ such that $\mathsf{Rev}_M(S_0) = W_M(S_0)$, so $\mathbb{E}_{\mathcal{A}}[\mathsf{Rev}_{\mathcal{A}}(S_0)] \geq \frac{1}{\ell}W(S_0) \geq \frac{1}{\ell}\varphi(S_0) > \frac{p/4}{\ell}W_{\mathcal{M}}(S)$. Let $H$ denote the event that $S_0$ is heavy. Then,

$$\mathbb{E}_{\mathcal{A}}[\mathbb{E}_{S_0}[\mathsf{Rev}_{\mathcal{A}}(S_0)]] = \mathbb{E}_{S_0}[\mathbb{E}_{\mathcal{A}}[\mathsf{Rev}_{\mathcal{A}}(S_0)]] \geq \mathbb{E}_{S_0}[\mathbb{E}_{\mathcal{A}}[\mathsf{Rev}_{\mathcal{A}}(S_0)|H]] \cdot \Pr(H) \geq \frac{p/4}{\ell}W(S) \cdot \Pr(H).$$

We now derive a lower bound on $\Pr(H)$. We have

$$\mathbb{E}_{S_0}[\varphi(S_0)] = \mathbb{E}_{S_0}\left[\frac{1}{n}\sum_{i=1}^{n} W_{\mathcal{M}}(S_0 \setminus \{i\})\right] = \mathbb{E}_{i \sim S}[\mathbb{E}_{S_0}[W_{\mathcal{M}}(S_0 \setminus \{i\})]] \geq \frac{p}{2}W_{\mathcal{M}}(S)$$

where in the final inequality we use the fact that $|S| \geq 2$ and that $W_{\mathcal{M}}$ is submodular, and so by Hartline et al. [20], $\mathbb{E}_{S_0}[W_{\mathcal{M}}(S_0)] \geq pW_{\mathcal{M}}(S)$. By Markov's inequality on the (nonnegative) random variable $W_{\mathcal{M}}(S) - \varphi(S_0)$,

$$\Pr(S_0 \text{ is heavy}) \geq \frac{(p/2)W_{\mathcal{M}}(S) - (p/4)W_{\mathcal{M}}(S)}{W_{\mathcal{M}}(S) - (p/4)W_{\mathcal{M}}(S)} = \frac{p/4}{1-p/4} \geq \frac{p}{4}.$$

Substituting this into our previous bound yields $\mathbb{E}_{\mathcal{A}}[\mathbb{E}_{S_0}[\mathsf{Rev}_{\mathcal{A}}(S_0)]] \geq \frac{p^2}{16\ell}W_{\mathcal{M}}(S)$, as desired. $\quad\square$

We now bound the number of heavy equivalence classes $\ell$. In order to do this, we introduce the notion of a *winner diagram*, which is a subgraph of the Hasse diagram of $S$. The winner diagram for $S$ is the following directed graph $\mathcal{G}$: each node is labeled $(S', \omega(S'))$ for some subset $S' \subset S$. The root node is labeled $(S, \omega(S))$. The children of node $(S', \omega(S'))$ are given by $(S' \setminus \{i\}, \omega(S' \setminus \{i\}))$ for each $i \in \omega(S')$. Figure 1 illustrates the winner diagram corresponding to a second-price auction for a single item with four bidders. Winner monotonicity will allow us to show that $\mathcal{G}$ contains a node that represents every equivalence class of $\equiv$.

**Lemma 4.2.** *$\mathcal{G}$ contains all equivalence classes of $\equiv$.*

*Proof.* Let $S^* \subseteq S$ be a set of bidders that arises as a winner set, that is, $S^* = \omega(S'')$ for some $S'' \supseteq S^*$. The set $S' \supset S'' \supset S^*$ is *maximal* for $S^*$ if $\omega(S') = S^*$ and $\omega(S' \cup \{i\}) \neq S^*$ for every $i \notin S'$. We show that for a given winner set of bidders $S^*$, there is a unique maximal set of bidders $S' \supseteq S^*$ such that $\omega(S') = S^*$. Initialize $S' = S^*$, and greedily add bidders from $S$ to $S'$ while $\omega(S') = S^*$ does not change. Due to winner monotonicity, if $i \notin \omega(S')$, then $i \notin \omega(S' \cup \{j\})$ for

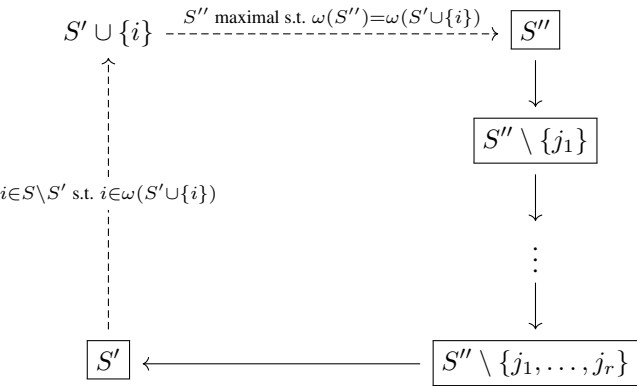

Figure 2: Illustration of the inductive step in Lemma 4.2. Boxed sets correspond to representative elements of equivalence classes in $\mathcal{G}$. Solid arrows represent directed edges in $\mathcal{G}$ from parent to child.

any bidder $j$. Hence, the order in which bidders are added by the greedy procedure does not matter, and therefore the final set $S'$ is the unique maximal set for $S^*$. Let the representative element of each equivalence class $[(S', \omega(S'))]$ be the one such that $S'$ is maximal for $\omega(S')$.

We prove the lemma by backwards induction on the size of the representative set $S'$ of any equivalence class. The base case of $|S'| = n$ is immediate since the root $(S, \omega(S))$ is the only node for which the representative set has size $n$. For the inductive step suppose that $\mathcal{G}$ contains a node for every equivalence class for which the representative set is of size at least $n'$. Let $(S', \omega(S'))$ be the representative of an equivalence class with $|S'| = n' - 1$. Let $i \notin S'$ be a bidder such that $i \in \omega(S' \cup \{i\})$. Such an $i$ exists due to winner monotonicity: if $i \in \omega(S)$, then $i \in \omega(S' \cup \{i\})$, since $S' \cup \{i\} \subset S$. Let $S''$ be the maximal set such that $\omega(S'') = \omega(S' \cup \{i\})$. We have $|S''| \geq |S' \cup \{i\}| > |S'|$, so by the induction hypothesis $\mathcal{G}$ contains a node labelled $(S'', \omega(S''))$. Now, there must exist a bidder $j_1 \in S'' \setminus S'$ such that $j_1 \in \omega(S'')$. If not, adding all the bidders in $S'' \setminus S'$ to $S'$ would not introduce any new winners, that is, $\omega(S' \cup (S'' \setminus S')) = \omega(S'') = \omega(S')$, which contradicts the maximality of $S'$ for $\omega(S')$. Therefore, the node $(S'', \omega(S''))$ has a child

$$(S'' \setminus \{j_1\}, \omega(S'' \setminus \{j_1\}))$$

($S'' \setminus \{j_1\}$ is maximal due to winner monotonicity). We may now find a bidder $j_2 \in S'' \setminus \{j_1\} \setminus S'$ such that $j_2 \in \omega(S'' \setminus \{j_1\} \setminus S')$ for the same reason as before. Continuing in this fashion yields a path from $(S'', \omega(S''))$ to $(S', \omega(S'))$, so $(S', \omega(S')) \in \mathcal{G}$, as desired. $\qquad \square$

Combining Lemmas 4.1 and 4.2 yields our main guarantee. Let $\gamma = \max_{S', i \in \omega(S')} \frac{\varphi(S' \setminus \{i\})}{\varphi(S')}$ and let $k = \max_{S'} |\omega(S')|$. We have $k \leq 2m$. The parameter $\gamma$ measures the smallest decrease in $\varphi$ between any two levels of $\mathcal{G}$, which we use to control the depth of nodes considered by our main mechanism $\mathcal{A}$. We stipulate that $\gamma < 1$. In Appendix C, we discuss how to remove this assumption and replace $\gamma$ with an appropriate parameter that is unconditionally strictly less than 1.

**Theorem 4.3.** *Let $|S| \geq 2$ and $S_0 \sim_p S$. We have*

$$\mathbb{E}_{\mathcal{A}}[\mathbb{E}_{S_0}[\mathsf{Rev}_{\mathcal{A}}(S_0)]] \geq \Omega\left(\frac{p^2}{k^{1+\log_{1/\gamma}(4/p)}}\right) W_{\mathcal{M}}(S).$$

*If $\mathcal{M}$ consists of revenue-monotonic mechanisms the slightly improved bound $\Omega(\frac{p^2}{k^{\log_{1/\gamma}(4/p)}})W_{\mathcal{M}}(S)$ holds. In particular, there exist mechanisms in $\mathcal{M}$ achieving the above guarantees in expectation.*

*Proof.* Let $\mathcal{G}'$ denote the restriction of the winner diagram $\mathcal{G}$ to nodes representing heavy equivalence classes. Each node of $\mathcal{G}'$ has out-degree at most $k = \max_{S'} |\omega(S')|$, and the depth of $\mathcal{G}'$ is at most $\log_{1/\gamma}\left(\frac{W_{\mathcal{M}}(S)}{(p/4)W_{\mathcal{M}}(S)}\right) = \log_{1/\gamma}(4/p)$ since $\varphi$ decreases by a factor of at least $\gamma$ when passing from a parent node to a child node (and $\mathcal{G}$ is truncated at nodes that are not heavy). Hence the number of nodes in $\mathcal{G}'$ is at most $k^{1+\log_{1/\gamma}(4/p)}$. If mechanisms in $\mathcal{M}$ are revenue monotonic, then we may modify $\mathcal{A}$ to randomize only over mechanisms corresponding to nodes of $\mathcal{G}'$ with out-degree 0. The

number of such nodes is at most $k^{\log_{1/\gamma}(4/p)}$. By Lemma 4.2, we may substitute this quantity for $\ell$ in Lemma 4.1, which completes the proof. $\qquad\square$

We have thus shown, via an application of the probabilistic method, that if $\mathcal{M}$ is a sufficiently rich mechanism class, there always exists $M \in \mathcal{M}$ that is robust to uncertainty in the market. Thus, $\sup_{M \in \mathcal{M}} \mathbb{E}[\mathsf{Rev}_M(S_0)] \geq \Omega(\frac{p^2}{k^{1+\log_{1/\gamma}(4/p)}})W_{\mathcal{M}}(S)$, with a slight improvement under revenue monotonicity.

In Section 3 we showed that if bidders can have allocational valuations there exist scenarios where only an exponentially small fraction of revenue can be guaranteed. Theorem 4.3 also holds in this setting under a stronger version of winner monotonicity, which we show in Appendix C.

## 4.1 Applications

The dependence of Theorem 4.3 on $\max_{S'} |\omega(S')|$ allows us to derive interesting families of guarantees when the seller places practical constraints on the auction setting. The first is a constraint on the number of winners, and the second is a bundling constraint that favors allocations that sell certain items together. Reasons for limiting the number of winners include: (1) avoiding the logistical hassle of having a large number of winners – this constraint is commonly used in sourcing auctions [21, 36, 37, 41] and (2) increasing competition to boost revenue, as is studied in Roughgarden et al. [34] (though in a different setting than ours). Kroer and Sandholm [25] show that even the vanilla VCG auction run with bundling constraints can yield significant revenue gains compared to VCG with no bundling constraints.

### 4.1.1 Limiting the number of winners

Suppose $\mathcal{M}$ is a class of mechanisms such that $|\mathsf{win}_M(S)| \leq n_0$ for all $M \in \mathcal{M}$ such that $W_{\mathcal{M}}$ is submodular, winner-monotonic, and satisfies the global-VCG-like property discussed previously. The proofs of all previous theorems go through with this constraint taken into account, with parameters modified correspondingly. Let $\varphi, \gamma$ be defined as previously.

**Theorem 4.4.** *Let $|S| \geq 2$ and $S_0 \sim p$, and let $\mathcal{M}$ be a class of mechanisms that sell to at most $n_0$ bidders. Then, there exists $M \in \mathcal{M}$ such that*

$$\mathbb{E}[\mathsf{Rev}_M(S_0)] \geq \Omega \left( \frac{p^2}{(2n_0)^{1+\log_{1/\gamma}(4/p)}} \right) W_{\mathcal{M}}(S).$$

In practical settings the auction designer might limit the number of bidders that can win a nonempty bundle of items – and in such cases $n_0$ can potentially be treated as a constant relative to $m$ and $n$.

In Appendix C, we consider VCG-like auctions that favor allocations that bundle certain items together. For such auctions, we obtain a guarantee in terms of the relevant bundling constraints.

## 4.2 General distribution over submarkets

So far we have stated our guarantees under the assumption that each bidder participates in the auction independently with probability $p$. When bidders participated independently with probability $p$, welfare submodularity was required to ensure that $\mathbb{E}[W_{\mathcal{M}}(S_0)] \geq pW_{\mathcal{M}}(S)$. In Appendix C we give a more general guarantee in terms of $\mathbb{E}[W_{\mathcal{M}}(S_0)]$ which only requires winner monotonicity.

## 5 How to choose an auction

Computing the mechanism $M \in \mathcal{M}$ that achieves the revenue guarantee of Theorem 4.3 can be accomplished by searching over the set $\{M_1, \ldots, M_\ell\}$ that $\mathcal{A}$ randomizes over, but this would potentially be a highly-inefficient procedure. Moreover, $\mathcal{A}$ itself is not computationally-efficient: determining the heavy sets of bidders, and determining the mechanisms $M_1, \ldots, M_\ell$ that are revenue maximizing for the heavy sets is an exhaustive procedure that would require enumerating over a potentially exponential number of subsets of $S$.

A more natural way for the mechanism designer to arrive at a mechanism is to learn from samples, which ensures that the mechanism designer uses the auction that (nearly) optimizes the expected preserved revenue, which could be significantly higher than what Theorem 4.3 guarantees.

We give a learning algorithm that the mechanism designer can use to learn a mechanism $\widetilde{M} \in \mathcal{M}$ that achieves an expected revenue of nearly $\sup_{M \in \mathcal{M}} \mathbb{E}[\mathsf{Rev}_M(S_0)]$. Our algorithm is similar in spirit to the learning-within-an-instance paradigm of Balcan et al. [5]. To describe the algorithm, we require the following structural notion of mechanism *delineability* introduced by Balcan et al. [3].

**Definition 5.1** (Balcan et al. [3])**.** Let $d, h \in \mathbb{N}$. A mechanism $M$ is $(d, h)$-*delineable* if (1) each mechanism $M \in \mathcal{M}$ can be parameterized by a $d$-dimensional vector $\theta \in \mathbb{R}^d$ and (2) for any set $S' \subseteq S$ of bidder valuations, there is a set $\mathcal{H}$ of at most $h$ hyperplanes in $\mathbb{R}^d$ such that $\mathsf{Rev}_\theta(S)$ is linear in $\theta$ over any given connected component of $\mathbb{R}^d \setminus \mathcal{H}$.

**Theorem 1.1.** *Let $\mathcal{M}$ be $(d, h)$-delineable class of mechanisms. A mechanism $\widetilde{M} \in \mathcal{M}$ such that*

$$\mathbb{E}[\mathsf{Rev}_{\widetilde{M}}(S_0)] \geq \Omega\left(\frac{p^2}{k^{1+\log_{1/\gamma}(4/p)}}\right) W_\mathcal{M}(S) - \varepsilon$$

*with probability at least $1 - \delta$ can be computed in $NhT + (Nh)^{O(d)}$ time, where $T$ is the time required to generate any given hyperplane witnessing delineability of any mechanism in $\mathcal{M}$ and $N = O\left(\frac{d \log(dh)}{\varepsilon^2} \log(\frac{1}{\delta})\right)$.*

*Proof.* We design a mechanism $\widetilde{M}$ such that $\mathbb{E}[\mathsf{Rev}_{\widetilde{M}}(S_0)] \geq \sup_{M \in \mathcal{M}} \mathbb{E}[\mathsf{Rev}_M(S_0)] - \varepsilon$ with high probability. The theorem statement then follows from Theorem 4.3. Our algorithm is based on the framework of empirical risk minimization from machine learning. The mechanism designer samples $S_1, \ldots, S_N \subseteq S$ independently and identically according to distribution $D$ on $2^S$. (We assume for simplicity that sampling according to $D$ can be done in a computationally efficient manner. If bidders participate independently with probability $p$, then the mechanism designer simply needs to flip $N$ coins of bias $p$ for each of the $n$ bidders in $S$.) The auction used will be the one that maximizes empirical revenue $\widetilde{M} = \mathrm{argmax}_{M \in \mathcal{M}} \frac{1}{N} \sum_{t=1}^N \mathsf{Rev}_M(S_t)$. Balcan et al. [3] show that $N = O\left(\frac{d \log(dh)}{\varepsilon^2} \log(\frac{1}{\delta})\right)$ samples suffice to guarantee that the expected revenue of $\widetilde{M}$ is $\varepsilon$-close to optimal with probability at least $1 - \delta$ over the draw of $S_1, \ldots, S_N$.

We now determine the computational complexity of maximizing empirical revenue. Our algorithm exploits similar geometric intuition that was used by Balcan et al. [3] to derive the above sample complexity guarantee. A similar approach has been used in other settings as well [4, 5].

For each $S_t \in \{S_1, \ldots, S_n\}$, let $\mathcal{H}_t$ denote the set of at most $h$ hyperplanes witnessing $(d, h)$-delineability of $\mathcal{M}$, and let $\mathcal{H} = \cup_t \mathcal{H}_t$, so $|\mathcal{H}| \leq Nh$. The number of connected components of $\mathbb{R}^d \setminus \mathcal{H}$ is at most $|\mathcal{H}|^d \leq (Nh)^d$. Each connected component is a convex polyhedron that is the intersection of at most $|\mathcal{H}|$ halfspaces. Representations of these regions as $0/1$ constraint-vectors of length $\mathcal{H}$ (a 0 in entry $h \in \mathcal{H}$ corresponds to one side of $h$, a 1 corresponds to the other side) can be computed in $\mathrm{poly}(|\mathcal{H}|^d)$ time using standard techniques [46]. Empirical revenue is linear as a function of $\theta$ in each connected component due to delineability, so the parameter $\theta$ that maximizes empirical revenue within a given component can be found by solving a linear program that involves $d$ variables and at most $|\mathcal{H}|$ constraints, which can be done in $\mathrm{poly}(|\mathcal{H}|, d)$ time. $\qquad\square$

Our algorithm has a run-time that is exponential in the number of parameters $d$ required to describe mechanisms in $\mathcal{M}$. In Appendix D, we study a class of sparse $\lambda$-auctions that can be described by a constant number of parameters. By leveraging practically-efficient routines for winner determination [35, 39, 40] (a generalization of the problem of computing welfare-maximizing allocations), we show how our empirical revenue maximization is computationally tractable for this setting (in particular, the run-time $T$ of computing the hyperplanes witnessing delineability is in terms of the run-time of winner determination).

# 6   Conclusions and future research

Our work is the first to formally study the problem of preserving revenue in a shrinking market. We gave a sample-based learning algorithm to design a mechanism that is robust to shrinkage

and uncertainty in the market. The crux of our analysis was a new combinatorial construction we introduced called a winner diagram. Our model of a shrinking market is simple and natural, and can serve as a testbed for many exciting mechanism design questions.

There are several open questions and new interesting research directions that stem from this study. The most immediate question is to derive tight bounds on revenue loss. There is a gap between the bound of Theorem 4.3 and the $(1 - p^2)$-fraction revenue loss of the simple example of a market with competition. Where does the true answer lie? Another interesting, and seemingly more difficult, setting is the one where the mechanism designer does not know the distribution $D$ over $2^S$ beforehand. Can he still arrive at an auction that is robust to the shrinking market? If the mechanism designer knows that each bidder participates independently with probability $p$, but does not know $p$, is it still possible to design a robust auction? Finally, we believe that the combinatorial bidder structure uncovered by our notion of a winner diagram could have interesting applications to other areas in mechanism design. While our analysis required a number of assumptions on the set of bidders, it would be interesting to extend the concept of a winner diagram to prove more general results with weaker assumptions. It would be interesting to extend our techniques to understand market shrinkage in other settings including objectives beyond revenue, other auction classes, and unlimited supply.

## Acknowledgments and Disclosure of Funding

This material is based on work supported by the National Science Foundation under grants CCF1733556, CCF-1910321, IIS-1901403, and SES-1919453, the Defense Advanced Research Projects Agency under cooperative agreement HR00112020003, an AWS Machine Learning Research Award, an Amazon Research Award, a Bloomberg Research Grant, and a Microsoft Research Faculty Fellowship. S. Prasad thanks Morgan McCarthy for interesting discussions about real-world shrinking (combinatorial) markets.

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
