# A Omitted results from Section 2

A $\lambda$-auction is an incentive-compatible parameterized generalization of the VCG auction where the mechanism designer may specify additive boosts to specific allocations. $\lambda$-auctions were introduced by Jehiel et al. [22]. The class of $\lambda$-auctions is a rich class of auctions and has been studied towards designing high-revenue combinatorial auctions [3, 38]. We use the notation $W(\alpha) = \sum_{i=1}^{n} v_i(\alpha)$, and $W(S) = \max_\alpha W(\alpha)$. Formally, a $\lambda$-auction run among $n$ buyers is specified by a vector $\lambda \in \mathbb{R}^{(n+1)^m}$ indexed by the $(n+1)^m$ possible allocations. The overall allocation chosen is

$$\alpha^* = \operatorname*{argmax}_{\alpha} W(\alpha) + \lambda(\alpha)$$

and bidder $i$ is charged a payment of

$$\max_{\alpha}(W_{-i}(\alpha) + \lambda(\alpha)) - (W_{-i}(\alpha^*) + \lambda(\alpha^*)).$$

An affine-maximizer auction [33] is a generalization of a $\lambda$-auction with multiplicative bidder-specific weights. We do not define them here, but all subsequent discussions on $\lambda$-auctions apply to affine maximizers as well.

The following proposition shows that $\lambda$-auctions are rich enough (and hence so are affine maximizers) to extract the entire social surplus as revenue if the bidders' valuations are known beforehand.

**Proposition A.1.** *For any set of bidders $S$, there exists a $\lambda$-auction, in the full-information setting with no incentive-compatibility constraints, with revenue equal to $W(S)$.*

*Proof.* Let $\alpha$ denote the efficient allocation among bidders in $S$ and let $\alpha_{-i}$ denote the efficient allocation among bidders in $S \setminus \{i\}$. We show that the $\lambda$-auction with $\lambda(\alpha) = 0$, $\lambda(\alpha_{-i}) = W(\alpha) - W_{-i}(\alpha_{-i})$, and $\lambda(\beta) = -\infty$ for all other allocations $\beta$ collects a payment of $v_i(\alpha)$ from each bidder, and thus extracts a revenue of $W(S)$. First, note that $\alpha$ is the overall allocation used since $W(\alpha) + \lambda(\alpha) \geq W(\alpha_{-i}) + \lambda(\alpha_{-i})$ for each $i$. To show that $\alpha_{-i}$ is the allocation used when bidder $i$ is absent, observe that

$$\begin{aligned}
W_{-i}(\alpha_{-i}) + \lambda(\alpha_{-i}) &= W(\alpha) \\
&\geq W(\alpha) - v_i(\alpha) \\
&= W_{-i}(\alpha) + \lambda(\alpha)
\end{aligned}$$

and

$$\begin{aligned}
W_{-i}(\alpha_{-i}) + \lambda(\alpha_{-i}) &= W(\alpha) \\
&\geq W(\alpha) - v_i(\alpha_{-j}) \\
&= W(\alpha) + v_j(\alpha_{-j}) - v_i(\alpha_{-j}) \\
&= W_{-i}(\alpha_{-j}) + W(\alpha) - W_{-j}(\alpha_{-j}) \\
&= W_{-i}(\alpha_{-j}) + \lambda(\alpha_{-j})
\end{aligned}$$

for any $j \neq i$. (We use the fact that if $\alpha$ allocates nothing to $i$, then $v_i(\alpha) = 0$.)

Thus the allocations used by this $\lambda$-auction are precisely the VCG allocations. The payment of bidder $i$ is therefore $(W_{-i}(\alpha_{-i}) + \lambda(\alpha_{-i})) - (W_{-i}(\alpha) + \lambda(\alpha)) = W(\alpha) - (W(\alpha) - v_i(\alpha)) = v_i(\alpha)$, and so the total revenue is $W(\alpha)$. $\qquad\square$

As a consequence of the above proof, $\lambda$-auctions and affine-maximizer auctions both satisfy the global-VCG-like property discussed in Section 2.

Since a variable group of bidders of variable size can participate in the $\lambda$-auctions we run, it is important to formalize how to distinguish between allocations since the $\lambda$-auction adds "boosts" to allocations specifically. We assume that the mechanism designer knows the valuations of the bidders in $S$ to begin with. So, each bidder can be thought of having an identity (for example, "the bidder who values apples at $x$ and oranges at $y$", or "the bidder with valuation function $v_4$"), and the mechanism designer knows the identities/valuations $v_1, \ldots, v_n$ of all bidders in $S$. An allocation, formally, is a mapping from items to bidder identities. Traditionally $\lambda$-auctions assume that the number of bidders is fixed, so allocations are usually interpreted as mappings from items to the position/index of a given

bidder. So, we, too, will consider $\lambda$-auctions run among all bidders, so an auction is parameterized by a $(n+1)^m$-dimensional vector that specifies boosts for allocations among all $n$ identities in $S$. However, our auctions also have to be well defined in the case of a shrinking market that has only a subset of the bidders. We address that as follows. If a $\lambda$-auction is run among a subset of bidder identities $S_0$ and chooses an allocation that allocates an item to a bidder identity not in $S_0$, we assume that the seller keeps that item.

## B  Omitted proof from Section 3

When bidder valuations can depend on what items the other bidders receive, we can construct a simple example where a random fraction of bidders participating incurs a much more significant revenue loss than the $(1 - p^2)$ fraction in our example for the combinatorial setting. Suppose there is a distinguished set of $k$ bidders (who can be viewed as unwealthy bidders) with negligibly low valuations for each bundle. All other bidders' (who can be viewed as wealthy bidders) valuation functions are defined to be above some threshold only on allocations that give a nonempty bundle to each of the $k$ distinguished bidders (and zero otherwise). In other words, while the wealthy bidders would like to receive items, they are not willing to participate unless unwealthy bidders are also guaranteed items. Then, for any nontrivial fraction of the revenue to be preserved, all distinguished bidders must participate in the auction, which occurs with probability $p^k$. Thus, any auction can preserve revenue at most $p^k \cdot W(S)$. The number of distinguished bidders $k$ can be taken to be as large as, for example, $m/2$. We now give a formal construction of the described example. Our construction satisfies the property that even the vanilla VCG auction extracts revenue nearly equal to the entire social surplus on the full set $S$ of bidders.

**Theorem B.1.** *For any $\varepsilon > 0$ there exists a set $S$ of bidders with allocational valuations such that*

$$\sup_{M \in \mathcal{M}} \mathbb{E}_{S_0 \sim_p S}[\mathsf{Rev}_M(S_0)] \leq p^{m/2} \cdot (\mathsf{Rev}_{VCG}(S) + 2\varepsilon) + \varepsilon$$

*for any auction class $\mathcal{M}$.*

*Proof.* For each item $1 \leq i \leq m/2$ we introduce two buyers with valuations $v_{i,1}, v_{i,2}$. For each item $m/2+1 \leq j \leq m$ we introduce a single buyer with valuation $v_j$. For $1 \leq i \leq m/2$ valuations $v_{i,1}$ are defined by $v_{i,1}(\alpha) = c$ if bidder $(i, 1)$ is allocated item $i$ and bidders $j = m/2+1, \ldots, m$ each receive at least one item, and $v_{i,1}(\alpha) = 0$ otherwise. Valuations $v_{i,2}$ are defined by $v_{i,2}(\alpha) = c - 2\varepsilon/m$ if bidder $(i, 2)$ is allocated item $i$ and bidders $j = m/2 + 1, \ldots, m$ each receive at least one item, and $v_{i,2}(\alpha) = 0$ otherwise. The only requirement on the valuations of bidders $j = m/2 + 1, \ldots, m$ is that $v_j(\alpha) \leq 2\varepsilon/m$ for all $\alpha$. The VCG auction would allocate item $i$ to bidder $(i, 1)$ for each $i = 1, \ldots, m/2$, and allocate the remaining $m/2$ items to bidders $j = m/2+1, \ldots, m$ such that each bidder $j$ receives exactly one item. The welfare of this (efficient) allocation is at most $cm/2 + \varepsilon$. The revenue obtained by VCG is at least $cm/2 - \varepsilon = W(S) - 2\varepsilon$. Let $S^*$ denote the set of small-valuation bidders $j = m/2 + 1, \ldots, m$. If each bidder shows up independently with probability $p$, the expected revenue of any auction $M$ is

$$\mathbb{E}[\mathsf{Rev}_M(S_0)] = \mathbb{E}[\mathsf{Rev}_M(S_0) \mid S^* \subseteq S_0] \cdot \Pr(S^* \subseteq S_0) + \mathbb{E}[\mathsf{Rev}_M(S_0) \mid S^* \nsubseteq S_0] \cdot \Pr(S^* \nsubseteq S_0)$$

$$\leq p^{m/2} \cdot \mathbb{E}[\mathsf{Rev}_M(S_0) \mid S^* \subseteq S_0] + \mathbb{E}[\mathsf{Rev}_M(S_0) \mid S^* \nsubseteq S_0]$$

$$\leq p^{m/2} \cdot \mathbb{E}[W(S_0) \mid S^* \subseteq S_0] + W(S^*)$$

$$\leq p^{m/2} \cdot W(S) + \varepsilon,$$

as desired. $\qquad\square$

## C  Omitted results and proofs from Section 4

### C.1  More details on $\gamma$

If $\gamma = 1$, we simply replace it with

$$\gamma := \max \left\{ \frac{\varphi(S' \setminus \{i\})}{\varphi(S')} : i \in \omega(S'), \frac{\varphi(S' \setminus \{i\})}{\varphi(S')} < 1 \right\} < 1$$

which is the worst-case non-trivial decrease in $\varphi$ across any two levels of the winner diagram. We are able to enforce that $\gamma < 1$ in this way because if $S'$ is such that $\varphi(S') = \varphi(S' \setminus \{i\})$, then the same mechanism achieves optimal revenue on both $S'$ and $S' \setminus \{i\}$. So these nodes can be considered jointly as a single node in the winner diagram, without incurring any penalty in the number of heavy equivalence classes $\mathcal{A}$ randomizes over.

## C.2 Allocational valuations

All of our results hold when bidders have allocational valuations. The only modification is that we require a stronger version of winner monotonicity, which states that if $i$ wins bundle $b$ under the mechanism that achieves $W_{\mathcal{M}}(S')$, then $i$ must win a bundle $b' \supseteq b$ under the mechanism that achieves $W_{\mathcal{M}}(S'')$ for any $i \in S'' \subseteq S'$. The weaker notion of winner monotonicity implies the stronger version when bidders have combinatorial valuations, but this is not necessarily the case when bidders have allocational valuations.

## C.3 Limiting the number of winners

*Proof of Theorem 4.4.* Given a set $S' \subseteq S$ of bidders, if a bidder in $\text{win}(S')$ is removed, at most one new bidder can win a nonempty bundle per the efficient allocation, due to winner monotonicity and the limit on the number of bidders. Thus, $\max_{S'} |\omega(S')| \leq 2n_0$. The remainder of the proof follows from the same arguments used to prove Theorem 4.3. $\square$

## C.4 Bundling constraints

A *bundling* is a partition of the set of items $\{1, \ldots, m\}$. An allocation $\alpha$ *respects* a bundling $\phi$ if no two items in the same bundle according to $\phi$ are allocated to different buyers. For a set of bundlings $\Phi$, the class of $\Phi$-*boosted $\lambda$-auctions* consists of all $\lambda$-auctions satisfying $\lambda(\alpha) \geq 0$ for all $\alpha$ that respects a bundling in $\Phi$ and $\lambda(\alpha) = 0$ otherwise. Let $W^{\Phi}(S)$ denote the maximum welfare of any allocation that respects a bundling in $\Phi$. Proposition A.1 holds with $W^{\Phi}(S)$ for the class of $\Phi$-boosted $\lambda$-auctions (the $\lambda$-auction constructed in Proposition A.1 can be shifted by a constant additive factor to make all boosts nonnegative). In the following theorem statement, $\varphi(S') = \frac{1}{n} \sum_{i=1}^{n} W^{\Phi}(S' \setminus \{i\})$.

**Theorem C.1.** *Let $\Phi$ be a set of bundlings. Let $S$ be a set of $n \geq 2$ bidders with valuations such that $W^{\Phi} : 2^S \to \mathbb{R}_{\geq 0}$ is submodular. Let $\gamma = \max_{S', i \in \omega(S')} \frac{\varphi(S' \setminus \{i\})}{\varphi(S')}$. Let $m_0$ be the greatest number of bundles in any bundling in $\Phi$. Let $\mathcal{M}$ be the class of $\Phi$-boosted $\lambda$-auctions. Then,*

$$\sup_{\lambda \in \mathcal{M}} \mathbb{E}_{S_0 \sim_p S}[\text{Rev}_\lambda(S_0)] \geq \Omega \left( \frac{p^2}{(2m_0)^{1 + \log_{1/\gamma}(4/p)}} \right) W^{\Phi}(S).$$

*Proof.* At most $m_0$ bidders can win a nonempty bundle of items, so $\max_{S'} |\omega(S')| \leq 2m_0$ by the same reasoning used to prove Theorem 4.4. The arguments used to prove Theorem 4.3 yield the desired bound. $\square$

## C.5 General distribution over submarkets

Our proof techniques easily generalize to handle any distribution $D$ over subsets of bidders since the only statistic of the distribution required is the expected welfare of a random subset of bidders $\mathbb{E}_{S_0 \sim_D S}[W_{\mathcal{M}}(S_0)]$. When bidders participated independently with probability $p$, submodularity of the welfare function was required to ensure that $\mathbb{E}[W_{\mathcal{M}}(S_0)] \geq p W_{\mathcal{M}}(S)$. In the following more general guarantee, which is in terms of $\mathbb{E}[W_{\mathcal{M}}(S_0)]$, we only need the more general condition of winner monotonicity.

**Theorem C.2.** *Let $S$ be a set of $n \geq 2$ bidders with valuations that satisfy winner monotonicity. Let $D$ be a distribution supported on $2^S$ with $\mathbb{E}_{S_0 \sim_D S}[W_{\mathcal{M}}(S_0)] = \mu \cdot W_{\mathcal{M}}(S)$. Let $\gamma = \max_{S', i \in \omega(S')} \frac{\varphi(S' \setminus \{i\})}{\varphi(S')}$ and let $k = \max_{S'} |\omega(S')|$. We have*

$$\sup_{M \in \mathcal{M}} \mathbb{E}_{S_0 \sim_D S}[\text{Rev}_M(S_0)] \geq \frac{\eta\mu}{k^{1 + \log_{1/\gamma}(1/\eta\mu)}} \left( \frac{\mu - 2\eta\mu}{2(1 - \eta\mu)} \right) \cdot W_{\mathcal{M}}(S)$$

*for all $0 \leq \eta \leq 1/2$.*

*Proof.* The proof is nearly identical to that of Theorem 4.3. The main modification is that $S' \subset S$ is heavy if $\varphi(S') \geq \eta\mu \cdot W_{\mathcal{M}}(S)$, and $\mathcal{A}$ randomizes over mechanisms corresponding to sets $S'$ with this property. Then, $\mathbb{E}_{S_0 \sim_D S}[\varphi(S_0)] \geq \frac{\mu}{2} W_{\mathcal{M}}(S)$ and so Markov's inequality yields $\Pr(S_0 \text{ is heavy}) \geq \frac{\mu/2-\eta}{1-\eta}$. The remainder of the proof is identical. □

Versions of Theorems 4.4 and Theorems C.1 for general distributions can be similarly obtained.

# D Omitted results from Section 5

Learning a high-revenue $\lambda$-auction would require a number of samples on the order of $(n+1)^m$ [3]. However, for sparse $\lambda$-auctions that are restricted to boost only a constant number of allocations, we can perform sample and computationally efficient learning while satisfying a similar guarantee to the ones derived for the entire class.

## D.1 $\lambda$ auctions with limited boosting

Let $\Gamma \subseteq \{1, \ldots, (n+1)^m\}$ be a set of allocations. The class of $\Gamma$-*boosted $\lambda$-auctions* consists of all $\lambda$-auctions satisfying $\lambda(\alpha) \geq 0$ for all $\alpha \in \Gamma$ and $\lambda(\alpha) = 0$ for all $\alpha \notin \Gamma$ (and can be specified by vectors in $\mathbb{R}^{|\Gamma|}$). $\Gamma$-boosted $\lambda$-auctions were introduced by Balcan, Sandholm, and Vitercik [3]. Let $W^{\Gamma}(S) = \max_{\alpha \in \Gamma} W(\alpha)$. We may derive a guarantee for this class of auctions analogous to Theorem 4.3. In the following theorem statement, $\varphi(S') = \frac{1}{n} \sum_{i=1}^{n} W^{\Gamma}(S' \setminus \{i\})$.

**Theorem D.1.** *Let $S$ be a set of $n \geq 2$ bidders with valuations such that $W^{\Gamma} : 2^S \to \mathbb{R}_{\geq 0}$ is submodular. Let $\gamma = \max_{S', i \in \omega(S')} \frac{\varphi(S' \setminus \{i\})}{\varphi(S')}$ and let $k$ be the maximum number of winners in any allocation in $\Gamma$. The class $\mathcal{M}$ of $\Gamma$-boosted $\lambda$-auctions satisfies*

$$\sup_{\lambda \in \mathcal{M}} \mathbb{E}_{S_0 \sim_p S}[\mathsf{Rev}_\lambda(S_0)] \geq \Omega\left(\frac{p^2}{(2k)^{1+\log_{1/\gamma}(4/p)}}\right) W^{\Gamma}(S).$$

## D.2 Algorithm for learning an auction from samples

We now give an algorithm that the mechanism designer can use to compute a $\Gamma$-boosted $\lambda$-auction that nearly achieves an expected revenue of $\sup_{\lambda \in \mathcal{M}} \mathbb{E}_{S_0 \sim_p S}[\mathsf{Rev}_\lambda(S_0)]$. Our algorithm leverages practically-efficient routines for solving winner determination, which is a generalization of the problem of computing welfare-maximizing allocations. The mechanism designer samples several subsets of bidders according to $D$, and computes the $\Gamma$-boosted $\lambda$-auction that maximizes empirical revenue over the samples.

While computing the empirical-revenue-maximizing auction is NP-hard in general, since winner determination is NP-hard, winner determination can be efficiently solved in practice [35, 39, 40]. Furthermore, when bidders have gross-substitutes valuations, winner determination can be solved in polynomial time [10]. The run-time of our algorithm is exponential only in $|\Gamma|$ but polynomial in all other problem parameters (including the run-time required to solve winner determination with $m$ items and $n$ bidders).

**Theorem D.2.** *Let $\mathcal{M}$ be the class of $\Gamma$-boosted $\lambda$-auctions. A $\hat{\lambda} \in \mathcal{M}$ such that*

$$\mathbb{E}_{S_0 \sim S}[\mathsf{Rev}_{\hat{\lambda}}(S_0)] \geq \sup_{\lambda \in \mathcal{M}} \mathbb{E}_{S_0 \sim S}[\mathsf{Rev}_\lambda(S_0)] - \varepsilon$$

*with probability at least $1 - \delta$ can be computed in $N(\min\{m, n\} + 1)w(m, n) + (Nn|\Gamma|)^{O(|\Gamma|)}$ time, where $w(m, n)$ is the time required to solve winner determination for $n$ buyers with valuations over $m$ items and $N = O\left(\frac{|\Gamma|\ln(n|\Gamma|)}{\varepsilon^2}\ln(\frac{1}{\delta})\right)$.*

*Proof.* The algorithm and proof are similar to Theorem 1.1. $\mathcal{M}$ is $(|\Gamma|, O(n|\Gamma|^2))$-delineable [3] Now, we explicitly describe how to generate the hyperplanes witnessing delineability. For each $1 \leq t \leq N$ let $\alpha^t$ denote the efficient allocation among bidders in $S_t$. For each $1 \leq t \leq N$ and each $i \in \mathsf{win}(S_t)$ let $\alpha^t_{-i}$ denote the efficient allocation among bidders in $S_t \setminus \{i\}$. For $i \notin \mathsf{win}(S_t)$,

$\alpha^t_{-i} = \alpha^t$. Determining these allocations requires at most $N + N \cdot \min\{m, n\}$ calls to the winner determination routine, since $|\text{win}(S_t)| \leq m$. The allocation used by any $\Gamma$-boosted $\lambda$-auction on $S_t$ is in $\Gamma \cup \{\alpha^t\}$, and the allocation used to determine the payment by bidder $i \in S_t$ by any $\Gamma$-boosted $\lambda$-auction on $S_t$ is in $\Gamma \cup \{\alpha^t_{-i}\}$.

For each $t$ and each pair of allocations $\alpha, \alpha' \in \Gamma \cup \{\alpha^t\}$, let $H(t, \alpha, \alpha')$ denote the hyperplane

$$\sum_{i \in S_t} v_i(\alpha) + \lambda(\alpha) = \sum_{i \in S_t} v_i(\alpha') + \lambda(\alpha').$$

For each $t$, each $i \in S_t$, and each pair of allocations $\alpha, \alpha' \in \Gamma \cup \{\alpha^t_{-i}\}$, let let $H_{-i}(t, \alpha, \alpha')$ denote the hyperplane

$$\sum_{j \in S_t \setminus \{i\}} v_j(\alpha) + \lambda(\alpha) = \sum_{j \in S_t \setminus \{i\}} v_j(\alpha') + \lambda(\alpha').$$

Let $\mathcal{H}$ denote the collection of these hyperplanes. The total number of such hyperplanes is at most $N(|\Gamma| + 1)^2 + Nn(|\Gamma| + 1)^2$. It is a basic combinatorial fact that $\mathcal{H}$ partitions $\mathbb{R}^{|\Gamma|}$ into at most $|\mathcal{H}|^{|\Gamma|} \leq (N(n + 1)(|\Gamma| + 1)^2)^{|\Gamma|}$ regions. Each region is a convex polytope that is the intersection of at most $|\mathcal{H}|$ halfspaces. Representations of these regions as $0/1$ constraint-vectors of length $|\mathcal{H}|$ can be computed in $poly(|\mathcal{H}|^{|\Gamma|})$ time using standard techniques [46]. Empirical revenue is linear as a function of $\lambda$ in each region, since the allocations used by $\lambda$ are constant within as $\lambda$ varies in a given region. Thus, the auction maximizes empirical revenue within a given region can be found by solving a linear program that involves $|\Gamma|$ variables and at most $|\mathcal{H}|$ constraints, which can be done in $poly(|\mathcal{H}|, |\Gamma|)$ time. □

One special case is when all allocations in $\Gamma$ are given the *same* boost. Then, the parameter space is $\mathbb{R}$, the number of relevant regions (subinteverals of $\mathbb{R}$) is $O(Nn|\Gamma|^2)$, and the algorithm in Theorem 1.1 has a run-time of $O(N \min\{m, n\}w(m, n) + Nn|\Gamma|^2)$. Mixed bundling auctions [22, 44] are an instance of this with $|\Gamma| = n$.

### D.3 Structural revenue maximization

Let $\Gamma_1 \subset \Gamma_2$ be collections of allocations, and let $\mathcal{M}_1$ and $\mathcal{M}_2$ denote the classes of $\Gamma_1$-boosted $\lambda$-auctions and $\Gamma_2$-boosted $\lambda$-auctions, respectively. Suppose the mechanism designer has drawn some number of samples $N$, and observes that the empirical-revenue-maximizing auction $\lambda_2$ over $\mathcal{M}_2$ yields slightly higher revenue than the empirical-revenue-maximizing auction $\lambda_1$ over $\mathcal{M}_1$, but $\lambda_2$ assigns nonzero boosts to significantly more allocations than $\lambda_1$ and is much more complex to describe. $\mathcal{M}_2$ is a richer auction class than $\mathcal{M}_1$, so it always yields higher empirical revenue, but there is the risk that it overfits to the samples. *Structural revenue maximization* allows the mechanism designer to precisely choose between such auctions by quantifying the tradeoff between empirical revenue maximization and overfitting [2, 3]. Instead of choosing $\lambda_2$ by default, the mechanism designer should choose $\lambda_k$, $k \in \{1, 2\}$ that maximizes empirical revenue minus a regularization term $\varepsilon_{\mathcal{M}_k}(N, \delta)$. The correct regularization term is precisely the error term

$$\varepsilon_{\mathcal{M}_k}(N, \delta) = O\left(\frac{|\Gamma_k| \ln(n|\Gamma_k|)}{\varepsilon^2} \ln\left(\frac{1}{\delta}\right)\right)$$

in the generalization guarantee for $\Gamma$-boosted $\lambda$-auctions obtained by Balcan, Sandholm, and Vitercik [3], which is fine-tuned to the intrinsic complexity of the auction class. Structural revenue maximization can be especially useful to the mechanism designer when there is a limit on the number of samples he can draw (due to a run-time constraint, for example). In this case, he may run the exact same geometric algorithm given in Theorem 1.1 with the modified objective of empirical revenue minus the regularizer described above. In particular, the algorithm may be run over the entire class of $\lambda$-auctions, and the mechanism designer effectively learns the best set $\Gamma$ of allocations to boost in order to guarantee high expected revenue while also generalizing well with high confidence.