# OpenReview forum: "Maximizing Revenue under Market Shrinkage and Market Uncertainty"
_NeurIPS.cc/2022/Conference — NeurIPS 2022 Accept_

### Official Review · Reviewer_jXoY · 2022-07-11

**Rating:** 6
**Confidence:** 3
**Soundness:** 3 good
**Presentation:** 3 good
**Contribution:** 2 fair

**Summary:**

This paper studies the mechanism design problem when all the players' valuation functions are known, and each player participates with certain probability. It provides a randomized mechanism with revenue lower bound guarantee and also a sample-based mechanism which can be implemented efficiently.

**Questions:**

Theorem 4.3 states there must exists a (deterministic) mechanism. Where does it prove its existence?


**Limitations:**

Not applicable.

**Strengths And Weaknesses:**

Strengths:
1. I like the idea that each player participates with uncertainty.
2. This paper provides a first lower bound result for the "market shrinkage" problem, and provides a sample-based algorithm.

Weaknesses:
1. The assumption that the valuations for all players are known is kind of restricted.
2. This paper seems more related to market uncertainty, but less to market shrinkage. What is the formal economic definition of market shrinkage?

---

> ### Author Response · Authors · 2022-08-02
> **Response to Reviewer jXoY**
>
> Thank you for your review! We respond to your specific comments and questions below.
>
> > “The assumption that the valuations for all players are known is kind of restricted.”
>
> We agree that removing the assumption that the bidders’ valuations are known is an interesting direction for future research. But, even under this assumption, we show in Section 3 that revenue loss can be drastic.
>
> > “This paper seems more related to market uncertainty, but less to market shrinkage. What is the formal economic definition of market shrinkage?”
>
> The market shrinkage aspect is due to the fact that only a fraction of the entire population of bidders participates in the market. The market uncertainty aspect comes from this fraction being unknown (each bidder participates independently with probability $p$, but the valuations of the bidders in the shrunken market is unknown to the mechanism designer).
>
> In Section 1.2 we provide a number of references to studies on shrinking markets. However, to the best of our knowledge, we are the first to propose a formal model of market shrinkage in multi-item settings, and we prove the first guarantees for how much revenue can be preserved in a shrinking market. Dobzinski and Uziely [12] studied revenue loss for a specific example of a shrinking market with a single item for sale where a single bidder drops out.
>
> > “Theorem 4.3 states there must exists a (deterministic) mechanism. Where does it prove its existence?”
>
> This conclusion is due to the probabilistic method. In Theorem 4.3 we prove that the randomized mechanism $\mathcal{A}$ achieves our revenue guarantee in expectation. Since $\mathcal{A}$ is a randomization over mechanisms in $\mathcal{M}$, there must exist at least one mechanism in $\mathcal{M}$ that achieves as much revenue as the expected revenue of $\mathcal{A}$. We will make this clearer.

---

> > ### Comment · Reviewer_jXoY · 2022-08-08
> > **Thanks for your response.**
> >
> > Thanks for your response. It addressed my concerns.

---

### Official Review · Reviewer_jPw2 · 2022-07-11

**Rating:** 6
**Confidence:** 1
**Soundness:** 3 good
**Presentation:** 3 good
**Contribution:** 3 good

**Summary:**

This paper models shrinking markets with uncertain size in multi-item setting, and proposes a sample-based learning algorithm with provable guarantees on how much revenue can be preserved in the face of uncertain market. The major novelty is the construction of a winner diagram, which captures all possible executions of an auction on an uncertain set of bidders, and a general possibility result by analyzing the winner diagram shows the proposed bound on the revenue guarantee.

**Questions:**

This paper is not in my area, and I have very limited related knowledge to give technical comments.
1. One question on bidder participation distribution: In Appendix C.5, the paper states that the proof techniques easily generalize to handle any distribution $D$ over subsets of bidders, does it imply that bidder participation probabilities can be mutually dependent? What is the impact on the revenue guarantee if the bidder participation distributions are not independent?
2. Do we have existing revenue guarantee on markets with known size? If yes, given the proposed model of market with uncertain size, how much loss do we encounter on revenue guarantee? Does the proposed algorithm have orderly optimal performance on revenue guarantee?

**Limitations:**

I did not notice any.

**Strengths And Weaknesses:**

Strength:
* The paper is well-written, all the ideas are clearly discussed and properly organized.
* The paper states that the proposed model of shrinking markets is the first formal model in multi-item setting, which seems technically non-trivial.

Weaknesses: NA

---

> ### Author Response · Authors · 2022-08-02
> **Response to Reviewer jPw2**
>
> Thank you for your review! We respond to your questions below.
>
> > “In Appendix C.5, the paper states that the proof techniques easily generalize to handle any distribution D over subsets of bidders, does it imply that bidder participation probabilities can be mutually dependent? What is the impact on the revenue guarantee if the bidder participation distributions are not independent?”
>
> Our result (Theorem C.2) for general distributions over subsets of bidders does not assume anything about independent participation, so bidder participation can certainly be correlated. Studying the more fine-grained impacts of correlated bidder participation on revenue is a very interesting question for future research.
>
> > "Do we have existing revenue guarantee on markets with known size? If yes, given the proposed model of market with uncertain size, how much loss do we encounter on revenue guarantee? Does the proposed algorithm have orderly optimal performance on revenue guarantee?"
>
> To the best of our knowledge, the setting where the size of the shrunken market is known has not been previously studied. Theorem C.2 in our paper would yield revenue guarantees for settings where the submarket is drawn according to some distribution over submarkets of fixed size.

---

> > ### Comment · Reviewer_jPw2 · 2022-08-07
> > **Rebuttal Reply**
> >
> > Thanks for the detailed reply! All my questions are addressed.

---

### Official Review · Reviewer_2dG7 · 2022-07-11

**Rating:** 6
**Confidence:** 3
**Soundness:** 4 excellent
**Presentation:** 3 good
**Contribution:** 4 excellent

**Summary:**

This paper studies revenue guarantees for multi-item auctions under shrinking markets, where the participation of bidders in auctions are independent and random. The paper first presents a motivating example for VCG auctions that demonstrate only an exponentially small fraction of revenue can be guaranteed under shrinking markets when bidders’ values also depend on what others receive. The paper then develops a probabilistic argument to show for a sufficiently large class of mechanisms, there must exist a mechanism that is robust to market shrinkage, and preserves a certain level of revenue. Finally, the paper presents a sample based approach to find a mechanism that yields revenue at least the presented guarantee on revenue with high probability.


**Questions:**

See above in strengths and weaknesses.

**Strengths And Weaknesses:**

Strengths:
The paper is well-written, and to the best of my knowledge, the probabilistic existence result for mechanisms that achieve certain revenue guarantees (along with the construction and analyses of the “winner diagram”) under shrinking markets is novel, and provides quite substantial contribution to the understanding of auction revenue in this particular market scenario. I find the motivating example in Section 3 that states in certain scenarios VCG can only retain an exponentially small fraction of revenue quite enlightening, and I appreciate the exemplifying numerics for revenue loss presented after Theorem 3.1. Despite being a theoretical paper, I feel the paper presents nice insights into challenges for auction design and revenue maximization in practice.

Weaknesses:
I do not see major weaknesses in the paper. Perhaps one question/concern I have for the overall methodology of mechanism $\mathcal{A}$ and the sample based approach to find a particular mechanism with good revenue guarantees, is that both rely on the knowledge of $p$  which is the probability that a bidder participates in an auction. If there is misspecification in this parameter (especially over estimations), it seems that certain mechanism equivalence classes may be ruled out which in the worst case may be those who correspond to the highest revenues. If this is true, is there any intuition regarding how sensitive are the revenue guarantees to misspecification of $p$? I may be missing something here, and would be great if the authors can clarify.

---

> ### Author Response · Authors · 2022-08-02
> **Response to Reviewer 2dG7**
>
> Thank you for your review! We respond to your question below.
>
> > “Perhaps one question/concern I have for the overall methodology of mechanism A and the sample based approach to find a particular mechanism with good revenue guarantees, is that both rely on the knowledge of p which is the probability that a bidder participates in an auction. If there is misspecification in this parameter (especially over estimations), it seems that certain mechanism equivalence classes may be ruled out which in the worst case may be those who correspond to the highest revenues. If this is true, is there any intuition regarding how sensitive are the revenue guarantees to misspecification of p? I may be missing something here, and would be great if the authors can clarify.”
>
> This is a very interesting question. If p is known approximately, e.g. if the mechanism designer believes that bidders participate independently with probability p’, where |p - p’| is sufficiently small, our results should still apply. Here is a sketch of such an argument:
>
> For each mechanism $M\in\mathcal{M}$, the function $f_M(p) = \mathbb{E}_{S_0\sim_p S}[Rev_M(S_0)]$ is a continuous function of $p$ (which can be seen by expanding the expectation as a sum with a term corresponding to each possible subset $S_0\subseteq S$). So for $p’$ sufficiently close to $p$, the expected revenues of the optimal mechanisms corresponding to $p$ and $p’$ can be made arbitrarily close. Now, consider running our sample-based learning mechanism in Theorem 1.1 where samples are drawn according to $p’$ rather than $p$. By Balcan et al. [3], the revenue of the empirically optimal mechanism corresponding to (sufficiently many) samples drawn according to $p’$ nearly matches the revenue of the true optimal mechanism corresponding to $p’$. By the above discussion on continuity, this revenue is very close to the revenue of the true optimal mechanism corresponding to $p$.
>
> However, this argument does not quantitatively pin down the sensitivity of our algorithms in terms of $|p - p’|$, but perhaps Lipschitz-type properties of the mechanism class could yield such a guarantee.
>
> Additionally, if no information about $p$ is known whatsoever (the mechanism designer does not even have access to a nearby $p’$), perhaps techniques from maximum likelihood estimation and/or distribution learning could be used in conjunction with our algorithms to derive guarantees in this setting.
>
> These are very interesting directions for future research!

---

### Official Review · Reviewer_3o7v · 2022-07-11

**Rating:** 7
**Confidence:** 3
**Soundness:** 3 good
**Presentation:** 4 excellent
**Contribution:** 4 excellent

**Summary:**

This paper studies the problem of maintaining a good revenue in a shrinking market for combinatorial auctions of n bidders and m items. The shrinking market is modeled as follows: every bidder n participates in the market independently with probability p (the valuations of the bidders participating in the market are hidden). The modeling of the problem is the first contribution of the paper. The second contribution is to show that in general settings, not handling the shrinking market specifically may not just incur a revenue that is p times the optimal revenue but significantly less. This result motivates even further the need to understand how much the revenue changes in a shrinking market. The third contribution is a possibility result, showing a lower bound on the revenue that can be extracted from a shrinking market. The result’s proof is based on a new construction, the winner diagram which may be of independent interest. The last contribution is a learning-from-samples-type algorithm for nearly achieving this revenue.


**Questions:**

See my question inside the first strength above about a more learning-theoretic phrasing of the problem. My other questions are mostly related to different assumptions in the model, so I listed them under “Additional Comments”.


**Limitations:**

properly addressed by the authors

**Strengths And Weaknesses:**

Strengths.
1) The problem studied in the paper seems fundamental and to the best of my knowledge, it has not been addressed in the literature. I was also thinking that although the paper is written in a way to emphasize the inherent MD component, the problem looks like a core ML theoretical problem phrased as follows: Let S be a sample on which an algorithm is trained and obtains a model M with loss L. If instead of S, the same algorithm is used on a subset of S (say S’ \subseteq S) then how far away are the two models M’ and M in terms of their losses? This is not precisely the question addressed in this work, but I was wondering whether there is a more ML-theoretic language to phrase the problem.
2) The paper is very well written. I also liked the story of the paper because I found it rather “complete”; for the problem at hand not only do the paper presents how much revenue can be preserved but also how to efficiently learn an auction achieving this revenue (i.e., an auction robust to market shrinkage or general market uncertainty).
3) The winner diagram technique (which is at the center of the construction for the possibility result) is quite neat. To be honest I was wondering whether there are certain classes of auctions where one can build and traverse these graphs fast enough so that we don’t have to learn from samples, but this is certainly not required for me to think positively about this paper.

Weaknesses.
Knowing p (or the distribution D) is restrictive. Do the current results hold even if you know p approximately? This would mean that there is a distribution D’ that is known with small distance from D. Such a result would strengthen the punchline of the paper even more.

Evaluation.
In my opinion, the weakness that I pointed out above was not as important as the strengths of the paper. Moreover, they mostly constitute interesting directions for future work. For that, I’m leaning positive for the paper.

Additional Comments.
* I understand that the assumption on the existence of independence in choosing the subset is done for technical reasons and I think it is okay given that this is one of the first studies of market shrinkage. But in reality shouldn’t we expect that there is some correlation between people who decide to leave the market? That is at least the case in all the examples mentioned in the introduction where there was an outside “force” pushing people out of the market.
* In lines 56-61: I would suggest adding a short description of what delineability is.
* In the first paragraph of Section 3, when describing v_j(b), would you mind explaining why this is correlated with i? And what is i in this case fixed to?
* After Def 5 I would suggest adding a short paragraph about what d, h, and \theta are for certain well-known examples, like the second price auction.

---

> ### Author Response · Authors · 2022-08-02
> **Response to Reviewer 3o7v**
>
> Thank you for your review! We respond to your comments and questions below.
>
> > “I was wondering whether there is a more ML-theoretic language to phrase the problem”
>
> We agree that this is a compelling direction for future research in learning theory and mechanism design.
>
> > “Knowing p (or the distribution D) is restrictive. Do the current results hold even if you know p approximately? This would mean that there is a distribution D’ that is known with small distance from D.”
>
> This is a very interesting question. If p is known approximately, e.g. if the mechanism designer believes that bidders participate independently with probability p’, where |p - p’| is sufficiently small, our results should still apply. Here is a sketch of such an argument:
>
> For each mechanism $M\in\mathcal{M}$, the function $f_M(p) = \mathbb{E}_{S_0\sim_p S}[Rev_M(S_0)]$ is a continuous function of $p$ (which can be seen by expanding the expectation as a sum with a term corresponding to each possible subset $S_0\subseteq S$). So for $p’$ sufficiently close to $p$, the expected revenues of the optimal mechanisms corresponding to $p$ and $p’$ can be made arbitrarily close. Now, consider running our sample-based learning mechanism in Theorem 1.1 where samples are drawn according to $p’$ rather than $p$. By Balcan et al. [3], the revenue of the empirically optimal mechanism corresponding to (sufficiently many) samples drawn according to $p’$ nearly matches the revenue of the true optimal mechanism corresponding to $p’$. By the above discussion on continuity, this revenue is very close to the revenue of the true optimal mechanism corresponding to $p$. This argument does not provide a quantitative guarantee in terms of $|p - p’|$, but perhaps Lipschitz-type properties of the mechanism class could yield such a guarantee.
>
> Additionally, if no information about $p$ is known whatsoever (the mechanism designer does not even have access to a nearby $p’$), perhaps techniques from maximum likelihood estimation and/or distribution learning could be used in conjunction with our algorithms to derive guarantees in this setting.
>
> Studying revenue guarantees when more general perturbations to the distribution $D$ are possible is a very interesting direction for future research, as we mention in Section 6.
>
> > “I understand that the assumption on the existence of independence in choosing the subset is done for technical reasons and I think it is okay given that this is one of the first studies of market shrinkage. But in reality shouldn’t we expect that there is some correlation between people who decide to leave the market?”
>
> Studying the effects of correlation in bidder dropout on revenue is a very interesting question for future research. In Appendix C.5 we provide a generalization of our theorem when the submarket of bidders is drawn according to a general distribution $D$ on $2^S$ (Theorem C.2). This result allows for correlation, but we agree that developing a tighter understanding here is a very interesting question.
>
> > “In lines 56-61: I would suggest adding a short description of what delineability is.”
>
> We will add this.
>
> > “In the first paragraph of Section 3, when describing v_j(b), would you mind explaining why this is correlated with i? And what is i in this case fixed to?”
>
> Thank you for catching this, and we apologize for the confusing typo. The phrase starting with “and bidder $j$ for $m + 1 \le j \le 2m$…” should instead read “and bidder $m + i$ for $1 \le i \le m$ has valuation $v_{m+i}(b) = c-\varepsilon/m$ if $i\in b$ and $v_{m+i}(b) = 0$ otherwise.” (There are $m$ items $\{1,\ldots,m\}$ and $2m$ bidders with valuation functions $v_1,\ldots,v_{2m}$. For each item $i$, there are two bidders who value $i$: the bidder with valuation function $v_i$ and the bidder with valuation function $v_{m+i}$.) We will make this clearer.
>
> > “After Def 5 I would suggest adding a short paragraph about what d, h, and \theta are for certain well-known examples, like the second price auction.”
>
> We will add such an example, specifically the class of second price auctions with reserve prices. For this class, $d = 1$, $h = 2$, and $\theta$ is the reserve price.

---

### Meta-Review · Area_Chair_n3Nf · 2022-08-21

**Recommendation:** Accept
**Confidence:** Certain

**Metareview:**

The reviews are all positive. The reviewers agree that the paper studies a fundamental problem with nice insights and interesting techniques, and the paper is well-written.

**Award:**

No

---

### Decision · Program_Chairs · 2022-09-14

Accept